# The Conceptual Analysis of Collaboration in the Occupational Therapy by Combining the Scoping Review Methodology

**DOI:** 10.3390/ijerph20116055

**Published:** 2023-06-05

**Authors:** Tatsunori Sawada, Kyongmi Oh, Mutsumi Namiki, Kounosuke Tomori, Kanta Ohno, Yuho Okita

**Affiliations:** 1Major of Occupational Therapy, Department of Rehabilitation, School of Health Sciences, Tokyo University of Technology, Tokyo 144-0051, Japan; tomoriks@stf.teu.ac.jp (K.T.); ohnoknt@stf.teu.ac.jp (K.O.); 2Department of Reha-Care, Funabashi Municipal Rehabilitation Hospital, Tokyo 273-0866, Japan; kyongmioh@gmail.com; 3Department of Rehabilitation, Gotanda Rehabilitation Hospital, Tokyo 141-0031, Japan; m623eight@gmail.com; 4Soaring Health Sports Wellness & Community Centre, Thomastown 3074, Australia; y.okita.griffith@gmail.com

**Keywords:** collaboration, occupational therapy, concept analysis, scoping review, goal-setting

## Abstract

Background: Collaboration is an important concept in goal-setting in occupational therapy. However, this concept is not stable due to various definitions. The purpose of this study was to clarify the concept of collaboration in occupational therapy. Method: A scoping review was used to search for all articles related to occupational therapy and collaboration. PubMed, Web of Science, CINAHL, and OT Seeker searches were conducted using predetermined keywords. Three examiners independently reviewed and assessed the quality of each study using Walker and Avant’s concept analysis method. Results: Results of the database searches yielded 1873 studies, 585 of which were deemed eligible to include in this review. Results showed five attributes (“active participation for the common objective”, “existence of something to share”, “matured communication and interaction”, “relationship founded on the respect and trust” and “complementing each other”) and two antecedents and several consequences. Conclusions: Our findings may contribute to collaborative goal-setting and occupational therapy.

## 1. Introduction

Collaboration plays a critical role in the goal-setting process, as supported by a substantial body of literature [1,2,3,4]. Collaborative goal setting is a process through which healthcare professionals and patients mutually agree upon health-related goals, which allows for personalized care that aligns with patient’s goals, values, and available resources [5]. Numerous studies have shown that engaging patients in goal-setting improves their confidence, motivation, and satisfaction with rehabilitation [6,7,8,9]. Additionally, healthcare professionals can benefit from enhanced patient participation and improved team dynamics that stem from a collaborative approach to goal-setting, ultimately leading to improved client outcomes [6,7,8,9].

Occupational therapy serves as an example of a profession that heavily relies on collaboration to deliver a client-centered approach [10]. The Canadian Model of Client-Centred Enablement (CMCE), a theoretical framework guiding the client–therapist relationship, highlights the importance of collaboration as one of the essential skills of an occupational therapist [11]. However, despite the recognized importance of collaboration in occupational therapy, the term itself has been defined in various ways within the field. For example, the American Occupational Therapy Association (AOTA) defines collaboration as working together with the mutual sharing of thoughts and ideas [12]. Bjørkedal et al. [13] described collaboration as power-sharing, involving acknowledgment, empathy, altruism, trust, and creative communication, in a joint intellectual effort to achieve a common goal. Another study defined collaboration as “working together towards a common goal” [14]. Hanna and Roger [15] presented several definitions of collaboration, including working together towards a common goal and a relationship based on cooperation rather than just an association between two or more people. However, despite these definitions, collaboration has been subject to interpretation and varied usage, often relying on expert opinions rather than distinct research methodologies. In the field of occupational therapy, collaboration typically refers to the relationships between the client and occupational therapist, between occupational therapists themselves, and between the team and other professions [11,12,15]. However, it is important to note that this concept may not be uniform across all contexts. Furthermore, the breadth, definition, and attributes of collaboration in occupational therapy remain unclear. Therefore, it can be argued that clearly establishing a definition for collaboration within the context of occupational therapy is crucial for advancing the field.

Given this situation, Emich [16] conducted a concept analysis of collaboration in nursing and identified various definitions. Although collaboration is an important concept in occupational therapy, particularly for goal-setting, no study has yet conducted a concept analysis of collaboration in occupational therapy. It can be argued that academically clarifying the concept attributes of collaboration, one of the most crucial concepts in occupational therapy, across all fields of occupational therapy would be useful. Therefore, the aim of this study is to develop a definition of the theoretical concept of collaboration using concept analysis and a scoping review.

## 2. Materials and Methods

### 2.1. Research Design

The methodology of Walker and Avants conceptual analysis was used to analyze the concept of “Collaboration”. As this method emphasizes exploring as many usages of the concept as possible, we employed the scoping review method, following the Preferred Reporting Items for Systematic reviews and Meta-Analyses extension for Scoping Reviews (PRISMA-ScR) Checklist (Appendix A) [17], to map the concepts holistically. A scoping review is a useful methodology to develop a comprehensive overview of existing evidence in a particular field. It can help explore and collate all information about the characteristics and application of the concept of collaboration in the field of occupational therapy to introduce future research needs. This scoping review followed Arksey and OMalley’s framework [18] which involves the five-step process as follows: (1) to identify the review research questions; (2) to identify the relevant studies; (3) to select studies; (4) to extraction data; (5) to summarize and report the results.

### 2.2. Review Questions

PRISMA-ScR generally uses patient, concept, context (PCC) to develop research questions, and we established the PCC as follows:

P: patient (the stakeholder of the collaboration with which an occupational therapist);

C: collaboration;

C: the report that is relevant to occupational therapy. The question for this study was, “What would be the concept of collaboration in the occupational therapy field”.

### 2.3. Inclusion Criteria and Search Strategy

Given the purposes of this study, any articles published in English that mentioned occupational therapy or occupational therapist were included in the analysis. Articles explaining the concepts related to this definition were included in the analysis. We included articles that were available from the library of Tokyo University of Technology library or the websites considering the fact that bulletin of associations and other sources can be a barrier for us to access the articles.

We searched the following databases up to 31 December 2019 for relevant publications: PubMed, Web of Science, CINAHL, and OT Seeker. Two key search terms, “collaboration” and “occupational therapy” were used for this search. The search formula was set as shown in Table 1 in PubMed, and other databases were searched accordingly.

Any discrepancies were resolved through discussion with other reviewers (Team 2: KT, YO, KO; 6th author).

### 2.4. Eligibility Criteria

The titles and abstract screening were completed independently by three authors (Team 1: TS, KO; 2nd author, MN), following the removal of duplicates. Then, the three authors independently screened the full-text papers and confirmed the eligible articles.

### 2.5. Data Extraction and Collation

A conceptual analysis was conducted on the selected literature. Walker and Avant’s approach to the conceptual analysis [19] was employed in this study, which is systematic and comprehensible method for clarifying the characteristics and attributes of a concept. This analysis consisted of eight steps (Table 2). The primary objective of this analysis was to clarify the concept of “collaboration” in the field of occupational therapy (Steps 1 and 2). All the available literature was reviewed to identify the various usages of the selected concepts (Step 3). The core of this method involved determining the attributes of the concept (Step 4), constructing a model case (Step 5), and developing additional case (Step 6). Once the defining attributes were established, cases that align with those attributes are created. After then, it was necessary to identify “antecedents and consequences” (Step 7) and “define empirical referents” (Step 8) that aligned with the identified attributes [19].

In this study, each researcher independently documented the results of the analysis for each article on the form (Appendix A). The data were carefully and iteratively reviewed by each member of Team 1 at each step (Steps 1–4). Starting from Step 4, the research team engaged in iterative discussions and made collaborative decisions. After determining the attributes, collaborative discussions within the team led to the development of a model case that encompassed these attributes (Step 5), as well as additional cases that exhibited partial or complete absence of these attributes (Step 6; borderline case, contrary case). Subsequently, the antecedents (preconditions preceding the concept) and consequences (outcomes) were thoroughly discussed and examined based on the literature obtained (Step 7). Finally, through the team’s discussions, empirical referents (Step 8) were established.

## 3. Results

A total of 1873 records were initially identified in our search, and 1533 individual articles were selected (Figure 1). About eight hundred resources were not able to be obtained from other countries because most of them included many foreign organization newsletters. Through the title and abstract screening, 927 studies were extracted, resulting in the inclusion of 606 studies. Through the full-text screening, we included 585 studies in the analysis. In this study, it was found that some of the references confirmed that the results overlapped or saturated upon analysis. As a result, only the key references have been included in the references section of this paper, while all other references are listed in Appendix A along with the results.

### 3.1. Literature Definitions of Collaboration (Step3)

#### 3.1.1. Dictionary Definitions of Collaboration

Collaboration is subject to different definitions across various dictionaries. Merriam-Webster’s Online Dictionary [20] provides three definitions: (1) working jointly with others, especially in an intellectual endeavor; (2) cooperating with or assisting an enemy of one’s country, particularly an occupying force; and (3) cooperating with an agency or instrumentality with which one is not immediately connected. The Cambridge Dictionary [21] defines collaboration as (1) the situation where two or more people work together to create or achieve the same thing; (2) the situation where people work with an enemy who has taken control of their country; and (3) the act of working together with other individuals or organizations to create or achieve something. In the Oxford Advanced Learner’s Dictionary [22], collaboration is described as (1) the act of working with another person or group to create or produce something; (2) a piece of work produced by two or more people or groups working together; and (3) the act of aiding the enemy during a war when they have taken control of one’s country. Therefore, collaboration can be understood as the collective work or production involving two or more individuals or groups. However, it is important to note that one of the definitions carries a negative connotation and specifically refers to aiding the enemy during wartime.

#### 3.1.2. Definition of Collaboration on Research

Collaboration is often used in research articles and projects. In this area, one definition is that “collaboration is a process in which people with common interests and goals pool their efforts for the purpose of accomplishing a specific project or task” [23]. Collaborative research is defined as a “research endeavour that pools the resources of any variety of researchers, agencies, scientists, clinicians, and representatives from different disciplines” [24]. Other studies have defined collaborative research as two or more individuals working together to investigate a research problem [25,26]. These definitions suggest that collaboration is sharing information, resources and working together to develop new outcomes or obtain new findings in research.

#### 3.1.3. Definition of Collaboration in Nursing Practice

In the nursing field, collaboration often refers to sharing goals, respecting the role of other professions, and working together with equality [27,28,29]. Collaboration is also applied to clients, with researchers clarifying that nursing collaboration with clients involves a close relationship, mutual understanding, and sharing goals based on interactive communication [30,31]. These concepts are based on interprofessional work and client-centered practice. Additionally, in a conceptual analysis study of nursing collaboration is defined as “an intraprofessional or interprofessional process by which nurses come together and form a team to solve a patient care issue or healthcare system problem with members of the team respectfully sharing knowledge and resources” [16]. This definition is similar to the concept of interprofessional work and education.

#### 3.1.4. Definition of Collaboration on Interprofessional Work (IPW)/Education (IPE)

Friend and Cook (2000) defined collaboration as “a style for direct interaction between at least two co-equal parties voluntarily engaged in shared decision-making as they work towards a common goal” [32]. Other researchers defined collaboration as “advancing a shared vision”, in which the positive view of the role of interdisciplinary work implies that opportunities such as the single assessment process will facilitate professionals’ desire to share and cooperate [33]. De Vries et al. [34] defined interprofessional collaboration as different health disciplines working together to solve patient problems; it is not just about coming together and working side by side (multi-disciplinary) but is the actual teamwork and problem-solving that occurs together (interprofessional). Anning et al. (2006) defined collaboration as two or more people committing to solve the same problem and having a clear sense of joint enterprise [35]. These definitions are all compatible with the concept of more than two people working together positively.

### 3.2. Attributes of Collaboration in Occupational Therapy (Step4)

According to Walker and Avant (2011), attributes are characteristics that repeatedly appear in a concept and help distinguish it from similar phenomena [19]. The results of analyzing 567 relevant resources helped identify five attributes associated with collaboration in occupational therapy, as shown in Table 3.

The first attribute is “active participation for the common objective”. Collaboration requires students to have the same learning objectives as well as active participation in classes, tasks, and problem-solving in the IPE [36,37,38]. The importance of this attribute in collaboration is also described in the IPW [39,40] and collaborative research [41,42]. This attribute highlights the importance of facilitating the active participation of team members in order to provide expected services to clients or to enable positive research outcomes through a collaboration process. More specifically, the client’s commitment is essential in delivering a client-centered practice through collaboration [13,43,44]. Further, Rebeiro [44] also stated that the client’s active participation is an essential element of collaboration following the concept of “true client-centeredness”. This attribute also applies to family members as clients in pediatrics and school-based occupational therapy areas, while collaboration cannot occur without a commitment from parents or teachers [15,45,46]. This attribute indicates the necessity of promoting stakeholders’ active participation and commitment toward the same goals and objectives through collaboration, which is a fundamental component of any collaboration.

The second attribute is the “existence of something to share”. Sharing in occupational therapy collaboration encompasses a wide variety of topics, such as vision [15,37,40,41], knowledge [39,42,47,48], information [39,48,49], goals [13,15,39,41,45,47,48,49], decision-making [15,45,47], responsibility [39,48,50], power [13,48], and experiences [15,48]. Sharing resources, such as ideas and funding, is also common in collaboration between researchers and institutions, [12,42,50]. In client-centered practice, sharing the client’s narrative [43], context [15], and values [51] is important.

The third attribute is “matured communication and interaction”. A considerable amount of research (465 articles) has identified communication and interaction as an attribute of collaboration. Collaboration requires active listening and open communication [52,53]. As open and interactive communication continues [49,54,55] it matures and becomes more creative [13,56]. Clients, students, and other professionals including researchers can be high-quality communication partners, requiring interactive communication through collaboration. [13,49,53,53,54,55,55,56]. If the interaction is used to promote the relationship between a product and an organization, the word “interaction” should be used rather than communication [57,58]. However, interaction does not necessarily occur in the collaboration process as the interaction itself can occur alone. This statement suggests that the term “interaction” needs to be used over “communication” when the relationship between a product and an organization is promoted through the collaboration process. This attribute is different from the next attribute in its different conceptualization of collaboration.

The fourth attribute is a “relationship founded on respect and trust”. Gee et al. [59] and Lidskog et al. [60] described mutual respect and trust as important factors for collaboration between each student. Rens and Joosten [61] introduced that respect and trust between occupational therapists and teachers are common characteristics of their collaboration. Several studies have explained that collaboration is based on mutual respect and trust between an occupational therapist and others [13,15,37,38,40,43,48]. This attribute indicates that an emotional foundation is needed in partnerships. Furthermore, respect and trust are suggested to be mutual, not one-way, which is similar to “matured communication and interaction” as above. However, this attribute is primarily concerned with the relationship associated with the emotional rather than the means.

The last attribute is “complementing each other”. Hansen, et al. [62] introduced that collaboration is a creative process that recognizes and respects the complementary skills and knowledge of each partner. Keough and Huebner [63] described that collaboration often involves working complementary between occupational therapists and psychologists to achieve a common goal. This attribute is often used in the IPE/IPW [39,40].

### 3.3. Construct a Model Case (Step 5)

Sayuri, an inpatient stroke survivor, was interviewed by an occupational therapist for medical services. During the initial interview, Sayuri shared her story of distress and anxiety about her difficulty looking after her children and household duties, and expressed her deep affection for her family. The occupational therapist actively listened to her story, understood the importance of family and meaningful work to her, and expressed empathy and gratitude for what she shared. Sayuri was impressed by the occupational therapist’s attitude and some suggestions (e.g., the occupational therapist suggested doing something special for Sayuri’s child’s upcoming birthday). Sayuri and the occupational therapist shared ideas with each other and they set a goal to make a birthday cake at home for her child’s birthday. The occupational therapist introduced Sayuri to use some self-help devices such as a non-slip ball and an easy-to-use knife, which she so excited to try out.

In this model case, Sayuri and the occupational therapist presented deep respect and trust in each other’s ideas, effective communication, and complementing each other’s skills and knowledge. They shared their ideas, knowledge, decision-making, and goals, and were actively committed to the occupational therapy process.

### 3.4. Construct an Additional Case (Step 6)

#### 3.4.1. Borderline Case

Yuji and Rena are a physiotherapist and an occupational therapist, respectively, who work together to assist a client with a fracture. They have a good working relationship and exchange their opinions on a daily basis to achieve their shared goal of helping the client live independently in the community. However, the collaboration between them lacks complementarity as they use the same techniques to improve joint range of motion, but their roles have significant overlap.

This is an example of a borderline case composed of characteristics of most of the attributes mentioned above, but one differs significantly. Although there are many attributes of collaboration such as participation, communication, interaction, and respect in this borderline case, there is a lack of complementary elements that is significant enough to make this a borderline case of collaboration.

#### 3.4.2. Related Case

A multidisciplinary team was providing services for a client with rheumatoid arthritis living at home. The physician orders the occupational therapist to provide occupational therapy to the client by home-visit, and the occupational therapist provided the services for one month and submits a written report to the physician on the progress.

This example of a related case shows some characteristics of the attributes mentioned but not all. This is a common situation in the home care field. Although the occupational therapist and physician are participating for a common purpose, the level of active participation and other factors are unclear, and the collaboration is still under developed.

#### 3.4.3. Contrary Case

Nami decided to cancel her occupational therapy services. Despite her desire to find a new job, the occupational therapist did not listen to her narrative but rather only imposed his own opinions. Further, the occupational therapist always did not provide the service for the expected session duration and left five minutes earlier, causing Nami to be alone to talk to other staff. This situation lasted for several weeks caused a break down of their working relationship and became a barrier for Nami to communicate with the occupational therapist.

This is an example of a contrary case that does not contain any attributes at all. Nami no longer wants to communicate with the occupational therapist. The relationship between the two has already collapsed and is the most distal to collaboration.

### 3.5. Antecedents and Consequences (Step 7)

We have identified two antecedents of collaboration in the field of occupational therapy. The first antecedent involves the presence of two or more objects with distinct aspects. Mattessich et al. [64] describe collaboration as a well-defined and mutually beneficial relationship between two or more organizations, aimed at achieving common goals. Vincent and Stewart [49] define collaborative consultation in occupational therapy as an interactive process involving multiple team members from various disciplines, working together to accomplish specific objectives [49]. Understanding these diverse objects is a fundamental requirement for initiating collaboration.

The second antecedent involves the presence of internal or external factors that require the involvement of multiple parties. Internal factors refer to the internal desire to cooperate with others prior to collaboration. Examples include clients or families expressing interest in occupational therapy [15,65], or researchers seeking the expertise of other professionals for their work [56,66]. External factors arise from situations where collaboration becomes necessary due to factors beyond one’s control. For instance, occupational therapy students are expected to collaborate as part of an education program that incorporates interprofessional education (IPE). Additionally, delivering interdisciplinary or multidisciplinary services in occupational therapy requires collaboration to ensure the quality of care. Under such circumstances, collaboration becomes a mandatory component driven by external factors [37,38,39,40]. Furthermore, considering occupational therapy’s client-centered practice and the inherent need for collaboration, the clinical practice of occupational therapy itself can be an external factor influencing collaboration [10,11,67].

Collaboration in occupational therapy has various consequences. On the positive side, collaboration enhances the quality of occupational therapy and healthcare services provided to clients [13,15,43,45,47,48,49]. Collaborative efforts also enable students in medical fields to acquire high-quality knowledge, skills, and experiences [36,37,38,50]. Nochajski [68] emphasizes that collaboration allows team members to learn from each other’s expertise, knowledge, and experiences. Collaborative research and projects yield new findings [66,69], programs [70], and products [71,72]. However, negative consequences also exist. Previous studies have revealed that occupational therapists perceive undesirable role overlap as an inevitable challenge within collaborative healthcare, leading to ambiguity regarding each profession’s distinct role [53,73]. Additionally, Savin-Baden et al. [74] highlight that collaboration among students can give rise to plagiarism, an unacceptable behavior within the occupational therapy profession.

In the overall context, no specific attributes related to previous prerequisites or outcomes were observed. The determination of attributes was based on how collaboration was utilized within the context of the paper.

### 3.6. Define Empirical Referents (Step8)

Empirical referments are measurable or observable behaviors that indicate a concept that has occurred and often corresponds to attributes [19]. In the context of collaboration in occupational therapy, the five attributes identified in this study serve as empirical referents. Of note, one study highlighted that collaboration means delivering a client-centered approach with goal-setting [75].

## 4. Discussion

Collaboration in occupational therapy has been defined in various ways leading to inconsistencies. This issue may be due to the lack of rigorous methodology used in previous studies. Therefore, we conducted a concept analysis and scoping review following Walker and Avant’s methodology to define collaboration in occupational therapy. Our results identified five attributes of collaboration: (1) Active participation for the common objective; (2) Existence of something to share; (3) Matured communication and interaction; (4) Relationship founded on respect and trust; (5) Complementing each other. These results represent the first study that defined collaboration in the occupational therapy field using a formal methodology comprehensively.

Compared to other health professions, occupational therapy collaboration places particular emphasis on working collaboratively with both clients and other healthcare professions. Our study revealed new attributes of collaboration, including “attribute 1—active participation for the common objective”, “attribute 3—matured communication and interaction” and “attribute 5—complementing each other”, whereas the attributes of collaboration in nursing were “sharing”, “teamwork”, and “respect” [16]. Furthermore, the attribute of “sharing” in the nursing study was often limited to IPW, but “sharing” identified in occupational therapy from our findings encompassed a broader meaning. Interestingly, our results did not identify “teamwork” as an attribute. Previous occupational therapy literature from Boshoff and Stewart [76] highlighted the difference between collaboration and teamwork. They emphasized the importance of separating the concept of these two as teams are expected to be formed through the application of collaboration [32,76]. Despite its importance, our results did not show a consistent definition of teamwork in the identified five attributes, suggesting that our results have a more comprehensive meaning due to our research objective and methodology as opposed to nursing. These results indicate that the focus of collaboration in nursing is on other professions, while the focus of collaboration in occupational therapy could be more comprehensive.

In contrast to previous research in the field of occupational therapy, our findings diverged in terms of the attributes associated with collaboration. While Gélinas [56] and Townsend et al. [77] highlighted the significance of “power sharing,” “*working with*”, “mutual respect, trust, and commitment”, “open communication”, “balanced power sharing”, “constant interaction”, and “shared vision, values, goals” as crucial elements of collaboration, our study identified additional attributes that are specific to occupational therapy. In occupational therapy, collaboration encompasses not only a wide range of knowledge but also decision-making and goal-setting, along with the importance of complementing one another (attribute 5). Our study emphasizes the concept of complementing each other’s deficiencies, which is vital for delivering client-centered occupational therapy practice and fostering effective collaboration with other professionals [62,63].

On the other hand, our comprehensive results can also shed light on the negative aspects of collaboration. Despite the generally positive perception of collaboration in occupational therapy, it can give rise to negative consequences, such as undesirable role overlap and plagiarism, emphasizing the need to address these negative aspects [29,53,73,74]. Moreover, it has been suggested that concept analysis should be conducted while considering our ethical values [19]. In our study, the researchers independently and iteratively analyzed the data over time and engaged in neutral discussions. Consequently, we elucidated that collaboration could yield several negative consequences through our analysis [53,73,74]. Therefore, we conclude that our findings are valid and useful for enhancing the understanding of collaboration in the context of occupational therapy.

The results of this study revealed three aspects of collaboration in occupational therapy. These include the relationship between clients and occupational therapists [10,11], the relationship between occupational therapists and other professionals [32,33,34,35], and the relationship between organizations and entities related to occupational therapy [70,71,72]. However, in this study, no consistent attributes were observed based on these aspects, as well as the preceding requirements and outcomes. In other words, it implies that attributes are not universally applicable across specific situations. Occupational therapy involves a transactional interplay among individuals, occupations, and the environment., highlighting the inherent complexity and diversity within the field. This complexity makes it challenging to categorize patterns under specific circumstances. Therefore, considering the complexity and diversity of occupational therapy, we believe that the results of this study are adequately justified and acceptable. Furthermore, this result shows that the diverse attributes of collaboration in the field of occupational therapy appear to be context-dependent, suggesting the need for an understanding tailored to specific situations rather than specific conditions.

## 5. Limitation and Future Study

One of the limitations of this review is that we included only articles written in English, suggesting that our findings may not cover the use of collaboration in all occupational therapy fields. The search process not only encompassed scholarly but also included diverse academic and professional organization newsletters, which are often challenging to access in Japan. Consequently, the inability to obtain over 800 papers can be considered as a limitation of this research. However, it is important to note that the study extensively analyzed approximately 600 sources, and we consider the findings to demonstrate sufficient theoretical saturation. Accordingly, our attributes are expected to be applicable in widespread areas of occupational therapy, from research to clinical practice, and we comprehensively investigated the concept of collaboration through a scoping review.

In the future, our findings can be useful for developing an assessment scale, utilizing collaboration in occupational therapy practice with clients and educational settings, and contributing to future studies.

## 6. Conclusions

This study provides a clear understanding of collaboration in occupational therapy. Our results suggest that the five attributes identified in this study are important for describing collaboration in occupational therapy. Defining the concept of collaboration in occupational therapy in this study can contribute to the improvement of occupational therapy practice and future studies.

## Figures and Tables

**Figure 1 ijerph-20-06055-f001:**
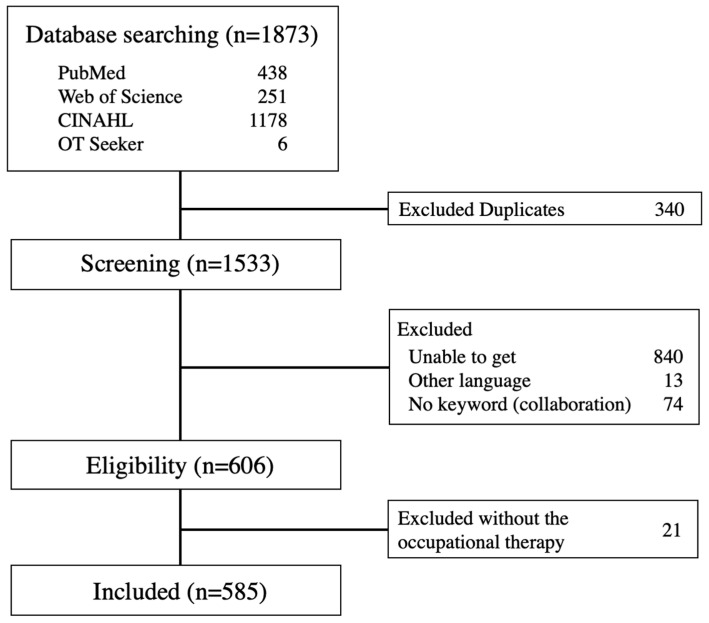
Scoping review process.

**Table 1 ijerph-20-06055-t001:** Search strategy in PubMed.

Search Formula	
#1	“Therapeutic Alliance” (Mesh) OR Collaboration
#2	“Occupational Therapy” (Mesh) OR “Occupational Therapy Department, Hospital” (Mesh)
#3	“Occupational therapy” AND “Collaboration”
#4	#1 AND #2 AND #3

**Table 2 ijerph-20-06055-t002:** Method of Walker and Avant’s concept analysis.

Steps	Explanation	Position
Step 1	Select a concept	Introduction
Step 2	Determine the purpose of the analysis	Method
Step 3	Identify all uses of the concept	Result
Step 4	Determine the defining attributes	Result
Step 5	Construct a model case	Result
Step 6	Construct an additional case	Result
Step 7	Identify antecedents and consequence	Result
Step 8	Define empirical referents	Result

**Table 3 ijerph-20-06055-t003:** Attributes of collaboration on occupational therapy.

Attribute Number	Contents of Attribute
Attribute 1	Active participation for the common objective (378 articles)
Attribute 2	Existence of something to share (398 articles)
Attribute 3	Matured communication and interaction (465 articles)
Attribute 4	Relationship founded on the respect and trust (196 articles)
Attribute 5	Complementing each other (46 articles)

## Data Availability

Raw data (Appendix A) are available https://zenodo.org/record/7751155#.ZBfwqOzP3SU (accessed on 20 May 2021), but some explanations are available in Japanese.

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
