# Peer review of "The Conceptual Analysis of Collaboration in the Occupational Therapy by Combining the Scoping Review Methodology"

_ijerph, 2023, doi:10.3390/ijerph20116055_

Round 1

Reviewer 1 Report

Dear authors,

The work is interesting and can bring clarity to the object of study in Occupational Therapy, however I have some suggestions: 

1. The concept of collaboration can be confusing. Therefore, it is necessary to introduce the subject with greater clarity. The introduction is too short.  

Collaboration in Occupational Therapy has two aspects: professional collaboration between therapists or patient collaboration in evaluation and treatment sessions. 

It is necessary that the type of collaboration you refer to is substantiated and explained in the introduction. 

2. The methodology must be clarified. The text is confusing. 

It is necessary to explain the steps and the bibliographic search in depth. 

I am surprised that you only use these databases. 

It is mandatory for Occupational Therapy to include google scholar. Traditionally, therapists have not written in indexed journals. Indexing in occupational therapy has a short history. 

In addition, you do not include in the search string all the synonyms to which the concept of 'collaboration' can refer. 

In order to properly substantiate and analyze this work, the search string must be improved. 

3. The results must be clear, concrete and objective. 

4. The discussion is poor. This topic allows for a larger and better quality discussion. 

5. Review the style of the references. 

I encourage you to reconstruct the work. It is a topic of interest within our discipline. 

Kind regards, 

Dear authors, 

English can be improved. 

Kind regards, 

Author Response

Dear reviewer 1

Thank you for your advice.

We response via Word file.

Reviewer 2 Report

Dear authors,

The manuscript is appropriate and of relevance to the occupational therapy community.

The introduction is adequate and the current state of the subject is shown. 

The material and method section is correct, although more detailed information on Walker and Avant's conceptual analysis methodology should be included. In addition, among the databases consulted, OTseeker appears, which stopped updating data in 2016, and the website itself recommends consulting other databases such as Pubmed and Pedro. Why was the research carried out on these databases and not others such as OTDbase?

In the screening of articles from the search performed, the reason why the articles could not be obtained should be indicated, as 840 articles are eliminated for this reason.

In the results section, it is recommended to include the articles or the number of articles that support each of the attributes, as well as to modify the expression "a considerable amount of research..." and indicate the amount involved. 

Best regards

Author Response

Dear Reviewer 2

Thank you for your revision.

We response via Word file.
